# Analysis of Septin 9 Gene Hypermethylation as Follow-Up Biomarker of Colorectal Cancer Patients after Curative Surgery

**DOI:** 10.3390/diagnostics12040993

**Published:** 2022-04-15

**Authors:** Miguel Leon Arellano, Mariano García-Arranz, Héctor Guadalajara, Rocío Olivera-Salazar, Teresa Valdes-Sanchez, Damián García-Olmo

**Affiliations:** 1Department of Surgery, Hospital Fundación Jimenez Diaz, 28040 Madrid, Spain; hector.guadalajara@quironsalud.es (H.G.); damian.garcia@uam.es (D.G.-O.); 2New Therapy Laboratory, Instituto de Investigación Sanitaria Fundación Jiménez Díaz, 28040 Madrid, Spain; mariano.garcia@quironsalud.es (M.G.-A.); olivera.rocio@hotmail.es (R.O.-S.); 3Bemygene, 46980 Valencia, Spain; teresa.valdes@bemygene.com

**Keywords:** colorectal cancer, liquid biopsy, Septin 9, free circulating DNA, predictive biomarker

## Abstract

Background: The Septin 9 test analyzes the methylation status of the SEPT9 gene, which appears to be hypermethylated in patients with colorectal cancer (CRC). This has been validated as a colorectal cancer screening test. Due to the high sensitivity and specificity found, the justification was to use it as a biomarker tool for monitoring minimal residual disease after radical surgery and recurrence. Methods: A prospective study was carried out at the Fundación Jiménez Díaz University Hospital extracting peripheral blood from 28 patients and 4 healthy donors. Free circulating DNA was obtained and subsequently a PCR reaction to quantify the number of methylated genes. Samples were obtained preoperatively and postoperatively at five to seven days, one and three months after surgery. Results: A total of 32 preoperative samples were analyzed. The sensitivity of the test to detect CRC was 55.6% and specificity was 100%. There were 22 postsurgical samples obtained at 5–7 days after surgery, the sensitivity to detect tumor recurrences was 100% and specificity was 75%. There were 21 samples analyzed 1 month after surgery exhibiting a sensitivity and specificity of 100% and 94.7%, respectively. At 3 months, 31 postsurgical samples were analyzed and the sensitivity and specificity were 66.7% and 80%. Conclusions: Detection of methylation of Septin 9 gene in circulating plasma DNA, obtained from a peripheral blood sample, may be a useful, non-invasive and effective method for detecting minimal residual disease and could therefore predict CRC tumor recurrences. The optimal time in our series to obtain the best prediction results based on Septin 9 methylation levels was one month after surgery. Despite these considerable findings, a study with more patients is necessary to obtain more robust conclusions.

## 1. Introduction

Colorectal cancer (CRC) is one of the most frequent tumors and one of the main causes of death from cancer in the world [1]. The incidence of CRC has grown in recent decades due to the establishment of screening programs and an increase in life expectancy [2]. Despite the development of screening programs, most patients are first diagnosed at the middle or late stage of CRC, leading to high mortality and poor prognosis. A surgical approach is the best treatment for patients with CRC, but recurrence due to persistence of minimal residual disease after resection is associated with a severe prognosis [3].

Approximately 25% to 40% of patients who undergo surgery with intent of curation for CRC will develop tumor recurrence within 5 years of cancer surgery [4]. An optimal surveillance protocol includes CT scans, colonoscopies, and measurement of tumor markers in serum, such as carcinoembryonic antigen (CEA), this last tool presenting a limited sensitivity and specificity. Normal CEA values can be found in almost 50% of cancers before surgical resection and often do not increase during recurrences [5]. The primary objective of a surveillance protocol of CRC patients is to improve survival rates [6].

To date, there lacks a specific tool with high sensitivity and specificity to detect minimal residual disease. This has led us to search for a precise, convenient and non-invasive biomarker that can detect the persistence or recurrence of CRC. In this study, we propose the methylation of the SEPT9 gene as a valuable tool.

The Septin 9 test analyzes the methylation status of the SEPT9 gene, which is hypermethylated in patients with CRC. The SEPT9 gene methylation assay, a blood-based test explicitly used for CRC detection and screening, was developed and used clinically in the last decade [7,8]. The hypermethylated SEPT9 gene has emerged as an accurate biomarker to detect CRC in peripheral blood and tumoral tissue [9,10].

Numerous validation studies analyzing the performance of the SEPT9 assay have been conducted, in particular, the FDA-approved the test Epi proColon 2.0, developed by Epigenomics AG, which reached sensitivity and specificity values ranging from 68% to 96% and from 80% to 97%, respectively [11,12].

We proposed to use this as a biomarker tool for detecting curative surgery and as a protocol for monitoring recurrence according to other studies [13,14]. If this biomarker becomes validated, we could offer better surveillance and a rational adjuvant therapy for patients in the future.

### Septin 9 as a Follow-Up Method

Although the Septin 9 test has been approved as a screening tool in CRC, an attempt has been made to check whether it may also have the potential to detect cases of recurrence and curative surgery. In a 2013 study with nine patients, Septin 9 methylation was detected before and after CRC surgery. Although the time at which the second measurement occurred was not determined, it was reported that the patients without methylation of Septin 9 were disease-free and those cases positive for methylation of Septin 9 presented with tumor recurrences [14]. Our research team in 2020 supported this possibility in a proof of concept with 10 patients, where 4 patients with recurrence of CRC had a 3-month postoperative blood test with positive hypermethylation of Septin 9 [15].

In other study from 2014, a five-year follow-up of patients undergoing CRC was performed detecting methylation of Septin 9 one year after surgery compared to postoperative CEA levels. This study concluded that Septin 9 methylation levels were more sensitive to detect tumor recurrences than CEA levels [13]. Further studies would help determine specifically which patients would benefit from adjuvant therapies, predict recurrences, metastasis, and survival based on Septin 9 results.

The goal of the present study was to determine if Septin 9 methylation test could be useful as a biomarker tool for monitoring minimal residual disease after radical surgery. The best time for this early analysis after surgery must also be determined so it may be used as an early-predictor of recurrence in patients undergoing curative surgery for CRC.

## 2. Materials and Methods

A prospective study was carried out with 28 patients diagnosed with CRC and 4 healthy donors enrolled in the study between May 2018 and June 2019 at the Fundación Jiménez Díaz University Hospital. The study was approved by the Clinical Research Ethics Committee. All patients signed the informed consent prior to the collection of the blood sample.

Inclusion criteria:Histopathological diagnosis of colorectal cancer;Surgical treatment of CRC, without the need for an ostomy;No prior treatment with chemotherapy or radiotherapy;Age range between 50–75 years;Informed consent signed.

All patients followed the usual CRC treatment and follow-up protocols according to the guidelines of our hospital. Demographic and clinical/radiological data were analyzed during follow-up.

Blood samples were obtained in four periods: preoperative, postoperative at four to seven days (hospital discharge), one month and three months after surgery (Figure 1).

### 2.1. Blood Samples

Ten milliliters of peripheral blood was collected from each patient in tubes with ethylenediaminetetraacetic acid (EDTA) Vacutainer^®^ processed to obtain plasma in less than 1 h after collection. The protocol for obtaining blood cell-free plasma was performed by double centrifugation at 1400× *g* for 12 min of the samples. The plasma obtained was stored in 3.5 mL aliquots at −80 °C for further processing.

### 2.2. Analysis of the Methylation Status of Circulating SEPT9 DNA in Plasma

Circulating free DNA (cfDNA) was obtained from 3.5 mL aliquots of frozen plasma from the included patients, and after quickly being thawed in a bath at 37 °C, it was processed according to the instructions of the manufacturer of the rapid kit of EpiproColon 2.0 plasma (Epigenomics AG, Berlin, Germany). Basically, this process consisted of treating the thawed DNA with ammonium bisulfite (EpiproColon Bisulfite Solution), which converts non-methylated cytosines into uracils. Bisulfite-processed DNA (bisDNA) was amplified by quantitative duplex PCR (qPCR) using the EpiproColon Sensitive PCR kit. This kit includes specific primers for the promoter region of the genes SEPT9 and ACTB (β-actin), the ACTB gene serving as an internal control to determine the concentration of DNA and adjust the parameters of the reaction.

The PCR reaction achieves the amplification of the heavy methyl radical, specifically blocking the unmethylated sequence converted with bisulfite and thus avoiding the amplification of unmethylated DNA. Subsequently a methylated SEPT9-specific fluorescent detection probe is combined to quantify the number of genes methylated [10].

Samples were amplified in triplicate using a 7500 Fast Real Time PCR System (Thermo Fisher Scientific, Waltham, MA, USA). EpiproColon positive and negative external controls were used in all independent runs.

PCR data for ACTB (β-actin gene) and methylated SEPT9 were recorded with 7500 FastDx software for each of the reactions in triplicate and then a cycle threshold (Ct) was generated for SEPT9 and ACTB within 45 cycles of amplification. The results were considered valid when the ACTB Ct was between 7.2 and 29.8, and the negative and positive external controls met the criteria specified by the manufacturer. A SEPT9 Ct value < 41.1 cycles was considered a positive result, while an indeterminate Ct was considered a negative result. Any other value of SEPT9 Ct was considered invalid.

### 2.3. Recurrence Criteria

Local recurrence or persistence was defined as clinical, radiological, or pathological evidence of the same type of histological tumor in the region of the anastomosis. Distant and regional recurrence or persistence was defined as clinical, radiological, or pathological evidence of spreading outside the site of the primary tumor.

For the analysis of the analytical and clinical results, two completely independent teams were created: on the one hand, the surgical group, which selected the patients and had access to the clinical history of each patient; and on the other hand, the laboratory group, which processed blood samples obtained from patients without access to clinical information. Once all of the tests were carried out, both groups met to analyze the results and draw conclusions of the study. To ensure the blind process of collection and handling of the samples, the results were revealed only after the collection and complete analysis of the patients.

## 3. Results

Twenty-eight patients were included in the study and four healthy controls. All patients met the inclusion criteria and signed an informed consent. A mean clinical-radiological follow-up of 36.7 months (26–57) was carried out on the series according to the conventional protocols used at the Fundación Jiménez Díaz University Hospital. The demographic and tumor characteristics are presented in Table 1. During the follow up, 6 patients presented tumor recurrences, which included 2 liver metastases at 6 and 12 months after surgery, 2 peritoneal carcinomatosis at 2 and 20 months after surgery, 1 pulmonary metastasis at 12 months after surgery, and another case of lymph node recurrence at 3 months after surgery.

### 3.1. Pre-Surgery Sample

Thirty-two preoperative samples were analyzed, out of 15 patients with a positive result to hypermethylation of Septin 9, it was confirmed that they matched with 15 cases of CRC. On the other hand, 13 samples with a negative result matched patients with a diagnosis of CRC and 4 negative results matched with the healthy controls. It is worth mentioning that after a clinical and pathological analysis, following the Amsterdam Criteria for hereditary nonpolyposis colorectal cancer, we found that one of the negative cases turned out to be a patient with Lynch Syndrome, which may explain the absence of methylation of Septin 9 in the plasma sample from this patient. The sensitivity of the Septin 9 methylation test to detect CRC in our series was 55.6%, and the specificity 100%. The positive predictive value (PPV) was 100% and negative predictive value (NPV) 23.5%.

### 3.2. Post-Surgical Sample at 5–7 Days

At 5–7 days after surgery, 22 samples were analyzed with 7 samples positive for hypermethylation, which matched with 2 patients who had tumor recurrence during follow-up. There were 15 samples that were negative, matching the 4 healthy controls and 11 patients who did not develop tumor recurrence. The sensitivity of the Septin 9 methylation test to detect cases of tumor recurrence 5–7 days after surgery was 100% in our series and the specificity was 75%. PPV was 28.6% and NPV was 100%.

### 3.3. Post-Surgical Sample at 1 Month

At 1 month after surgery, 21 samples were analyzed. Of them, 3 samples were positive, matching with 2 cases of tumor recurrence. There were 18 samples with negative results, which coincided with the 4 healthy controls and 14 disease-free patients. Test sensitivity at 1 month after surgery to detect cases of tumor recurrence was 100% and specificity was 94.7%, with a PPV 66.7% and a NPV of 100%.

### 3.4. Post-Surgical Sample at 3 Months

At 3 months after surgery, 31 samples were analyzed. Septin 9 hypermethylation was present in 9 samples, matching 4 patients with tumor recurrence. A total of 22 samples had a negative result, which 4 coincided with 4 healthy controls and with 16 patients without evidence of tumor recurrence. The sensitivity of the Septin 9 methylation test to detect cases of tumor recurrence 3 months after surgery was 66.7%, specificity 80%. PPV of 44.4% and NPV of 90.9%.

In our series, we found differences in the sensitivity and specificity depending on the time of collection and analysis of the samples. Preoperative samples provide us with information about the power of this test to detect cases of CRC as a screening test. The other ranges of time provided us with information of the usefulness of this test to detect cases of tumor recurrences. According to our series, the highest sensitivity and specificity of the Septin 9 test to detect tumor recurrence was 1 month after surgery (Table 2).

### 3.5. CEA with Tumor Recurrences

Measurement of CEA tumor marker was performed in 28 patients diagnosed with CRC 6 months after surgery. The objective was to determine whether in our series there was a relationship between elevation of tumor marker and the presence of tumor recurrences.

A total of five patients were found to have increased CEA levels at six months after surgery. On the other hand, 23 patients presented no elevation in CEA levels.

Within the group of 5 patients with elevated CEA levels (above 5.0 ng/mL), only 2 coincided with confirmed cases of tumor recurrence. In the group of 23 patients who did not present elevated CEA levels, 4 of them presented with tumor recurrence (Table 3 and Table 4).

## 4. Discussion

The present work has focused on the liquid biopsy method based on the methylation of the Septin 9 gene that is emerging as an accurate biomarker to detect CRC in peripheral blood. This method is currently accepted by the FDA as screening method of CRC in patients between 50 and 75 years old, as long as fecal or visual tests cannot be performed.

The Septin 9 test consists of the analysis of the methylation status of the gamma promoter region of the SEPT9 V2 gene transcription, which is methylated differentially in patients with CRC and has proven to be effective in the screening of CRC as a non-invasive and precise method [10,14,16]. This study exposes the usefulness of the Septin 9 methylation test in a peripheral blood sample for the diagnosis of CRC and has been proposed as a follow-up method to detect early CRC recurrences. It is important to note two important limitations: the age range of the patients for which this test has been validated as well as that there appears to be an effect on the results depending on the race of the patients. For example, the Korean study [14] did not obtain as favorable of results on sensitivity and specificity for the use of Septin 9 as compared to the German and North American studies [10,16].

Our research team published in 2020 as proof of concept, where plasma from 10 patients with CRC was collected preoperative and three months after surgery, with promising results. [15] We therefore enlarged the number of patients to confirm our results and proposed two extra-periods of earlier blood collection at five to seven days and one-month after surgery. Our hope was to verify if we could demonstrate an early-predictor of CRC recurrences and propose an optimal time, prior to the start of adjuvant treatments, to analyze the methylation of the SEPT9 gene.

One aspect to consider has been that despite the low number of healthy volunteers, the constancy of their results. Also, with the inclusion of one patient with HNPCC, this finding enabled us to test the laboratory process assuring blindness in each process. To assure validity and contrast the results of our trial, we developed a strategy with two independent teams. On the one hand, a clinical team, who selected the patients that met the inclusion criteria, collected, coded the samples, and had access to the information of the clinical evolution. On the other hand, a laboratory team, who processed the blood samples and generated the results of the methylation of the SEPT9 gene. This ensured a blind method by comparing the results of the tests with the clinical evolution of the patients.

### 4.1. Preoperative

To date, different studies have been published in which the sensitivity and specificity of Septin 9 methylation is measured for CRC screening, in some series sensitivity and specificity of 48.2% and 91.5%, respectively, was reported [9]. More recently in another study, a sensitivity and specificity of 73.3% and 81.5% respectively was reported [10]. However, these studies were conducted in German and North American populations. The results of Septin 9 methylation to detect CRC cases by screening in Asian populations were reported much lower compared to previous western series, with a sensitivity of 17% and specificity of 90.6% [14]. The overall sensitivity of Septin 9 methylation to detect CRC in our study was 55.6% and the specificity was 100%.

It seems however, that in our setting the effectiveness of Septin 9 methylation to detect CRC is slightly lower than that reported in Western series and higher than Asian series. This leads us to believe that the sample size should be increased to clearly define its validity for the population in our setting.

### 4.2. Postoperative

Overall, in our series, the result with the best sensitivity for predicting cases of CRC recurrence was one month after surgery, reaching a sensitivity and specificity of 100% and 94.7% respectively. These values were lower when we analyzed the samples taken five to seven days after surgery. We think that one-month after surgery allows the Septin 9 molecule to be eliminated from the body and thus provide better results compared to other periods of time. Therefore, one month after surgery, we are in a situation of stability in the measurement of Septin 9 methylation and, if confirmed in a trial with a larger number of patients, we can decide if adjuvant post-surgery treatments are necessary. There was no correlation between the type of recurrence and the hypermethylation level.

Results at three months after surgery had lower sensitivity to predict CRC recurrence compared to one month after surgery. It is very possible that the initiation of adjuvant chemotherapeutic treatments from the first month after surgery altered the value of gene methylation.

A previous study by Lee et al. [14] used a sample of 27 patients with CRC where 4 presented a recurrence, of which, 3 had a negative result of Septin 9 postoperatively, and only 1 presented a positive result. In this study, the postoperative collection time of the samples were not reported. Ma [17] reported that at three-month after CRC operation, there was a significant decrease in the Septin 9 methylated abundance, especially in stage III and distal cancers.

In our series, it is worth mentioning that the four healthy controls always had a negative result for Septin 9 methylation in peripheral blood. We believe this result should be confirmed in a future trial with a larger number of healthy volunteers. The results, as previously mentioned, to ensure the blind process of collection and manipulation of the samples, were revealed after a complete collection of patient’s data.

The proposed test appears to demonstrate a high success rate with the additional advantage of being a minimally invasive procedure that obtained results in less than 96 h.

We also consider it necessary to analyze the appropriate time frame after surgery in which we can detect hypermethylation of Septin9 DNA in plasma without losing prognostic value. In our study, we proposed three time periods to be able to successfully study this value.

In our series, the sensitivity of CEA tumor marker to detect tumor recurrence was 33% and a specificity was 86.4% making it was much lower than the measurement of Septin 9. Despite the fact, that the samples obtained from peripheral blood for the CEA measurements were performed six months after surgery coinciding with the protocols from our hospital.

### 4.3. Proposals for the Future from Methylation of Septin 9

Considering our results, we are certain that it would be beneficial to generate a larger multicenter study to further support the predictive capacity of Septin 9 methylation 1 month after CRC surgery. Extended results would provide more evidence in the value this marker contains in aiding decisions and perhaps adjust the clinical guidelines for CRC, avoiding unnecessary adjuvant treatments for many patients and lowering the cost of treatments.

In this regard, we believe that this test may be useful in other challenging clinical settings and not just as a follow-up test for early detection of CRC recurrence or as a preoperative screening result. Septin 9 methylation could be useful as an epigenetic biomarker to confirm a complete clinical response following neoadjuvant therapy in locally advanced rectal cancer, especially when adapted to a non-surgical “watch and wait” protocol [18]. This tool could be added to the radiological and clinical evaluation after a complete response. Further investigation is need, but the rationale behind our study indicates that this tool provides immense potential for the future care of patients with CRC and rectal cancer.

Another aspect that should be clarified in the future, is the age at which the sensitivity of SEPT9 gene methylation is best assessed considering that in the last decade there has been a decrease in the average age of CRC patients [19]. Finally, this test could be used as a measurement tool to confirm complete resection after oncological surgery for CRC peritoneal carcinomatosis or even after transanal microsurgery in early-stage rectal cancer to ensure complete excision and as a non-invasive follow-up. Our initial data are presented as the first non-invasive alternative to determine a complete or curative surgery in patients after undergoing treatment for CRC. We acknowledge, however, that we must increase the sample size in order to assert these proposals.

## 5. Conclusions

Methylation detection of the Septin 9 gene in circulating DNA in plasma, obtained from a peripheral blood sample, can be a useful, non-invasive, and effective method for screening for CRC in our environment.

In addition, methylation detection of Septin 9 gene in free DNA circulating in plasma, can predict tumor recurrences of CRC. In our series, the optimal time to obtain the best prediction results based on the levels of Septin 9 methylation was one month after surgery. While the samples studied at five to seven days and three months after surgery were less sensitive in detecting cases of CRC recurrence compared to the samples studied 1 month after surgery.

## Figures and Tables

**Figure 1 diagnostics-12-00993-f001:**
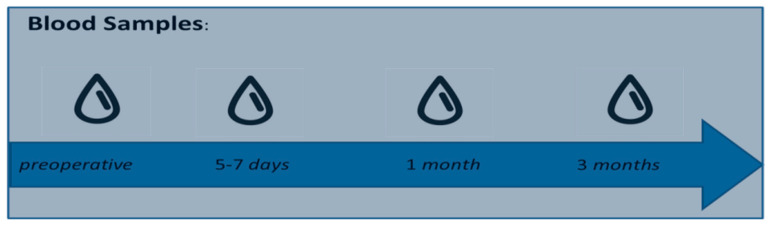
Period of time of blood samples.

**Table 1 diagnostics-12-00993-t001:** Demographics and tumor characteristics.

Variable	n = 28	
**Gender**	Male 16	
	Female 12	
**Age**	65.5 (52–77) years	
**Stage**	I	5
	II	9
	III	10
	IV	4
**Localization**	Right	16
	Left	3
	Sigmoid	7
	Rectum	2

**Table 2 diagnostics-12-00993-t002:** Sensitivity and specificity of the Septin 9 test.

	Preoperative	5–7 Days	1 Month	3 Months
**Sensitivity**	55.6%	100%	100%	66.7%
**Specificity**	100%	75%	94.7%	80%

**Table 3 diagnostics-12-00993-t003:** CEA values with tumor recurrences.

CEA Values	Patients	
**Elevated (5)**	2	Recurrence
	3	Disease free
**Not Elevated (23)**	4	Recurrence
	19	Disease free

**Table 4 diagnostics-12-00993-t004:** Comparative analysis CEA values and SEPT9 methylation.

Test	CEA (6-Month)	Septin 9 (1-Month)
**Sensitivity**	33%	100%
**Specificity**	86.4%	94.7%

## Data Availability

The data presented in this study are available in the article.

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
