# Peer review of "Analysis of Septin 9 Gene Hypermethylation as Follow-Up Biomarker of Colorectal Cancer Patients after Curative Surgery"

_diagnostics, 2022, doi:10.3390/diagnostics12040993_

Round 1

Reviewer 1 Report

It would be better to repeat the test after 6 months or one year after surgery to find the best results 

Author Response

Dear reviewer

Thank you for your comment and for reviewing our article.

Regarding your comment, we agree with you that repeating the test after 6 months or one year after the surgery, would give us more information. But we think that the objective of the study was to evaluate if we can use this test as an early-predictor of recurrence, we believe that these findings are much more interesting compared to a late-predictor of recurrence. Also we think that an early-predictor could, at some point, avoid adjuvant therapies and excessive follow-up tests.

Reviewer 2 Report

The Authors analyzed the methylation status of the SEPT9 gene, which is hypermethylated in colorecta cancer as a biomarker to monitor minimal residual disease and eventual recurrence in colorectal cancer operated on with radical surgery. The topic is of interest and the data meaningful.

  • Can you please explain why the timepoints for assessment were chosen?
  • Recurrence events were local or distant, as states in the material and methods. Did you not considered regional (nodal) recurrence? Why? In the results there was 1 nodal relapse. Please align M&M and results.
  • Table 2. I suggest to use a complete contingency table with absolute numbers, relative % and marginal values.
  • Is there a correlation between the type of recurrence and the probability of hypermethylation?
  • Can you please state the time of recurrences? Is it correlated with the hypermethylation status?

Author Response

Dear reviewer.

Thank you for your reviewing our article and for your valuable comments.

I will respond point-by-point: 

- Can you please explain why the timepoints for assessment were chosen?

Timepoints were chosen for checking if this test could be used as an early-predictor of recurrence. The first timepoint was 4-7 days after surgery, this match with the discharge day. The second timepoint was 1 month after surgery, this period match with the time before adjuvant therapies starts, and the third timepoint was 3 months, this period could at some point decrease the follow up tests performed to the patients. 

- Recurrence events were local or distant, as states in the material and methods. Did you not considered regional (nodal) recurrence? Why? In the results there was 1 nodal relapse. Please align M&M and results.We agree with you that we should consider regional recurrence. You can see the correction in Materials and Methods.

- Table 2. I suggest to use a complete contingency table with absolute numbers, relative % and marginal values.

It would be desirable to have marginal values and relative percentage, but the limited number of cases did not allowed us to make a more advanced statistic analysis. Also we believe that a complete contingency table would not be so "visual" to communicate the sensitivity and specificity of the 4 time-periods. Information of the number of patients  in each period is described in paragraphs 3.1, 3.2, 3.3 and 3.4.

- Is there a correlation between the type of recurrence and the probability of hypermethylation?Thank you for this comment, we did not find any correlation between the type of recurrence and the hypermethylation status level. We will add this valid information in Discussion

- Can you please state the time of recurrences? Is it correlated with the hypermethylation status?There was no correlation between the time of recurrence and the hypermethylation status. Thank you for your comment about the times of recurrences, we believe as you, that this information is important. You will see this data in the corrected paper.

Round 2

Reviewer 2 Report

I do not have any further comment.